# Efficacy and moderators of psychological interventions in treating subclinical symptoms of depression and preventing major depressive disorder onsets: protocol for an individual patient data meta-analysis of randomised controlled trials

David D Ebert,[1] Claudia Buntrock,[1] Jo Annika Reins,[2] Johannes Zimmermann,[3] Pim Cuijpers[4]

For numbered affiliations see end of article.

**Correspondence to**
Dr David D Ebert;
david.ebert@fau.de

## ABSTRACT

**Introduction** The long-term effectiveness of psychological interventions for the treatment of subthreshold depression and the prevention of depression is unclear and effects vary among subgroups of patients, indicating that not all patients profit from such interventions. Randomised clinical trials are mostly underpowered to examine adequately subgroups and moderator effects. The aim of the present study is, therefore, to examine the short-term and long-term as well as moderator effects of psychological interventions compared with control groups in adults with subthreshold depression on depressive symptom severity, treatment response, remission, symptom deterioration, quality of life, anxiety and the prevention of major depressive disorder (MDD) onsets on individual patient level and study level using an individual patient data meta-analysis approach.

**Methods and analysis** Systematic searches in PubMed, PsycINFO, Embase and the Cochrane Central Register of Controlled Trials were conducted. We will use the following types of outcome criteria: (A) onset of major depression; (B) time to major depression onset; (C) observer-reported and self-reported depressive symptom severity; (D) response; (E) remission; (F) symptom deterioration; (G) quality of life, (H) anxiety; and (I) suicidal thoughts and behaviours. Multilevel models with participants nested within studies will be used. Missing data will be handled using a joint modelling approach to multiple imputation. A number of sensitivity analyses will be conducted in order test the robustness of our findings.

**Ethics and dissemination** The investigators of the primary trials have obtained ethical approval for the data used in the present study and for sharing the data, if this was necessary, according to local requirements and was not covered from the initial ethic assessment. This study will summarise the available evidence on the short-term and long-term effectiveness of preventive psychological interventions for the treatment of subthreshold depression and prevention of MDD onset. Identification of subgroups of patients in which those interventions are most effective will guide the development of evidence-based personalised interventions for patients with subthreshold depression.

## Strengths and limitations of this study

► A strength of the presented individual patient data meta-analysis (IPD-MA) is that this approach allows sufficient statistical power to evaluate specific effects for specific kinds of treatments for patients with certain characteristics, in order to select the best possible treatment for an individual patient (ie, personalised medicine).

► One limitation of the IPD-MA is that while investigating moderators of treatment outcome, one very much relies on the variables that have been assessed in the primary studies. However, many of the relevant predictors and moderators associated with depression onset or differential treatment response reported in the literature were not assessed in the included studies.

► Another limitation of the IPD-MA approach is that some bias is introduced because not all eligible trials can be included in the analyses due to author non-response, lack of ethical approval to share the data or that data are not available anymore.

**PROSPERO registration number** CRD42017058585.

## INTRODUCTION

Major depressive disorder (MDD) is highly prevalent,[1–4] associated with substantial impairment[5 6] and economic costs.[7–9] Psychological treatments have been shown to be effective in the treatment of depression.[10 11] However, it has been estimated that even under the hypothetical scenario of full coverage with and adherence to evidence-based treatments, approximately only one-third of the disease burden attributable to MDD could be averted.[12] Moreover, in practice, the majority

of depressed people remain untreated,[3][13] even in high-income countries.[14][15]

Therefore, attention has increasingly been shifted to the prevention of MDD onsets.[16][17] One specific form of prevention is indicated prevention. In such interventions, subthreshold symptoms are treated in order to prevent the transition to a full-blown depressive disorder.[17] Meta-analytic evidence shows that indicated psychological preventive approaches can be effective in preventing depressive episodes.[18] The latest systematic review, which included randomised trials that have been published up to March 2012, found psychological interventions for subclinical symptoms to be effective in reducing the risk of developing an MDD at 6-month (incidence rate ratio (IRR)=0.61; 5 studies) and 12-month follow-up (IRR=0.74; 4 studies). Since then, many more randomised controlled trials (RCT) have been published, warranting an update of the evidence.

Moreover, the treatment of subclinical symptoms of depression itself is relevant. Subthreshold depressive symptoms are highly prevalent,[19] related to increased mortality,[20] poorer quality of life,[21] increased healthcare service utilisation[22] and vast economic costs.[23] However, results for the treatment of subclinical symptoms are yet conflicting. Pharmacological interventions are unlikely to have a clinical advantage over placebos in treating subthreshold depression.[24] In addition, although a recent meta-analysis found small-to-moderate effect sizes for psychological interventions on depressive symptom severity at post-treatment compared with usual care,[25] four studies using clinician-rated outcomes did not indicate significant positive results.[25] Moreover, we are not aware of any systematic review exploring the long-term effects of treatments for subclinical symptoms with regard to depressive symptom severity, and effects on other relevant outcomes such as anxiety or quality of life have not been examined.

Another issue not yet addressed is the possibility that the effectiveness of psychological interventions for subthreshold depression varies across patients and not all subgroups of patients profit from such interventions. Given that the number of people from specific subgroups is often small in single trials, and randomised trials are usually powered to detect overall treatment effects, RCTs are mostly underpowered to perform adequately subgroup and moderator analyses.[26] As studies also seldom report effectiveness for different patient characteristics, it is impossible to examine patient-level moderators using traditional meta-analytic approaches.

Individual participant data meta-analyses (IPD-MA) can overcome some of the limitations of the conventional meta-analyses on study level.[27–29] By pooling the primary data of individual trials, it is possible to conduct analyses not reported in original studies and obtain large enough sample sizes with sufficient power to examine the effects in relevant subgroups and identify outcome moderators.[30]

The present study aims to examine the short-term and long-term effects of psychological interventions compared with control groups in adults with subthreshold depression on depressive symptom severity, treatment response, remission, symptom deterioration, quality of life, anxiety and the prevention of MDD onsets using an IPD-MA approach. Moderators on individual patient level (eg, sociodemographic, clinical characteristics) and study level (eg, type of treatment delivery, number of sessions, theoretical basis) on intervention outcome will be explored in the pooled data set. In addition, we will analyse the intervention effects and moderators of effects in specific subgroups of interest (eg, using only data from patients with low education, chronic medical conditions, and so on).

## METHOD

### General study approach

First, a systematic review is performed to identify eligible papers. Corresponding authors of selected studies will be contacted and asked to provide raw data from their studies. The current study will be completed in compliance with the Preferred Reporting Items for Systematic Reviews and Meta-Analyses statement. IPD will be aggregated and a priori defined moderator variables will be analysed using a multilevel model approach.

### Eligibility criteria

In this IPD-MA, we will (A) include randomised trials in which (B) the effects of a psychological treatment (delivered individually, in a group-based, bibliotherapy, internet-based format) were compared with a comparison group (waiting list, care as usual, psychological placebo, pill placebo, antidepressant medication) (C) in adults (D) with clinically relevant depressive symptoms (E) but no MDD at baseline, (F) assessed with a standardised diagnostic interview (see below) to exclude participants with full-blown mood disorder at baseline. Psychological interventions are defined as the application of psychological mechanisms and interpersonal stances derived from psychological principles for the purpose of assisting people to modify their behaviours, cognitions, emotions and/or other personal characteristics in directions that the participants deem desirable.[31][32] Clinically relevant depressive symptoms will be defined as scoring above a cut-off score on a self-rating depression questionnaire; scoring above a cut-off score on a clinician-rated instrument; or meeting criteria for minor depression according to the Diagnostic and Statistical Manual of Mental Disorders, or the International Classification of Diseases. We will also include studies in which participants with a diagnosed depressive disorder were examined and we will then exclude participants with a full-blown disorder on an individual basis using the primary data. No language restrictions will be applied.

### Types of outcome measures

We will use the following types of outcome criteria: (A) onset of MDD; (B) time to MDD onset; (C)

observer-reported and self-reported depressive symptom severity; (D) response; (E) remission; (F) symptom deterioration; (G) quality of life; (H) anxiety; and (I) suicidal thoughts and behaviour. MDD will be assessed with clinical interviews such as the Structured Clinical Interview for DSM Disorders,[33] the Composite International Diagnostic Interview,[34] or the Mini-International Neuropsychiatric Interview.[35] Depressive symptom severity will be measured using standardised depression outcome measures such as the Beck Depression Inventory,[36] Hamilton Depression Rating Scale,[37] or the Center for Epidemiological Depression Scale.[38] If both observer-rated and self-report measures are available, we will explore intervention effects on both outcome measure types. If several observer-rated or self-report measures are used, preference will be given to the mostly used measures across the different studies in order to increase comparability. If the type of outcome measures varies between studies, these measures will be transformed into standardised scores (using the common metric approach[39] or, if this is not possible, z-transformation). We will also dichotomise scores on depressive symptoms to explore the effects on two response criteria (a 50% reduction in symptoms for relative change; a minimum absolute change in symptoms according to the Reliable Change Index)[40] and remission (scoring below a predefined cut-off score). Symptom deterioration rates will be calculated using a predefined absolute worsening of symptoms from baseline to follow-up using the Reliable Change Index[39] and 50% symptom increase. Quality of life will be transformed to quality-adjusted life-years using, if possible, the British value set for EQ-5D-3L utility values[41] and Brazier's algorithm for SF-6D utility values,[42] respectively. Anxiety severity will be measured using standardised self-report measures, such as the Hospital Anxiety and Depression Scale[43] or Beck Anxiety Inventory.[44] Note that we are planning to reduce the complexity for moderator analyses by only focusing on (A) onset of MDD and (C) depressive symptom severity.

## Moderators

We will investigate both moderators on individual patient level (eg, sociodemographic, clinical characteristics) as well as on study level (eg, type of treatment delivery, number of sessions, theoretical basis). Published papers are examined to identify potential moderators on patient level that have been assessed across studies. We will explore variables that have shown to predict differential treatment outcome in psychological treatments for depression[45 46] and variables that are associated with depression onset.[47–49]

Clinical and personality characteristics that shall be investigated include depressive symptom severity,[50] lifetime history of MDD,[51 52] number of previous depressive episodes,[51 53] anxiety symptoms,[51] comorbid mental health disorder (eg, anxiety disorder),[52] previous exposure to depression treatment, family history of common mental health disorders,[52 54 55] global assessment of functioning, sleeping problems,[56–58] neuroticism,[50]

recent life stress,[59] childhood adversities,[55] traumatic events,[60] significant life events (in the previous year),[61 62] daily hassles, emotion regulation,[63] poor self-perceived health (quality of life),[51 56 62] self-esteem,[64–66] (chronic) medical conditions,[57 58 67] physical functioning/disability,[56] mastery, worrying, body mass index, rumination, interpersonal problems,[53 62] body dissatisfaction,[66 68] physical activity level,[56 69] diet quality,[69] alcohol/substance use,[52 56 62] smoking,[56 67] resilience,[70] social support/integration[52 57 63 66] and perceived social rejection/mobbing. Sociodemographic variables that shall be examined include sex,[54 67 71 72] age,[52 71] education,[58 73] marital status,[73] relationship status,[71] living alone,[55] employment,[55] ethnicity (minority status),[74] economic deprivation/poverty[57 62 74] and parenthood (motherhood).[67] It is expected that not all studies that will be included will assess all variables. Hence, a precondition for including a variable as a moderator in the actual analyses is availability of sufficient data. Intervention characteristics that will be examined include the intervention format (individual, group or guided self-help), the number of treatment sessions, overall treatment duration, session frequency,[75] the type of delivery (internet, face-to-face), the control condition (placebo/attention control, care as usual, waitlist, alternative treatment), type of psychotherapy (cognitive behavioural therapy, problem-solving, interpersonal or other type) and study quality.

## Timing of outcome assessments

All postintervention assessments will be pooled and treated as one assessment, despite varying time frames in included studies. Treatment duration will be controlled for if found to be associated with the dependent variable. We expect varying follow-up periods of the studies and will therefore categorise follow-ups into meaningful categories, such as follow-up that occurred 3–7 months (follow-up I), 8–13 months (follow-up II) or over 14 months (follow-up III) after baseline.

## Searches and study selection

For the identification of potential studies for inclusion, we will use a database of papers on the psychological treatment of depression described in detail elsewhere.[76] For this database, studies have been identified from PubMed, PsycINFO, Embase and the Cochrane Central Register of Controlled Trials. In addition, previous meta-analyses of treatments for depression were screened for this database to ensure that no randomised trial was missed. These searches identified a total of 16 407 abstracts (12 196 after the removal of duplicates); from this, 1885 full-text papers of RCTs on treatments for depression were retrieved for possible inclusion in the database. These papers will be screened for inclusion in this meta-analysis. A further literature search will be conducted for studies published since the last update of the database (studies published up to December 2017 will be considered for inclusion). In addition, relevant authors in the field of depression

prevention will be asked whether they are aware of any yet unpublished study that might fit the inclusion criteria.

Corresponding authors will be contacted for each of the identified papers and will be asked to provide raw data from their study. If an author does not respond after 1 month, a second attempt to contact him/her will be made. If the second contact fails, another author of the study will be contacted and invited to participate. A second attempt to contact this author will follow a month later if no response is received, and so forth, until a maximum of three authors were contacted. Study data will be considered unavailable in the event that no study author has responded to multiple contact attempts or if all contacted authors indicate that they no longer have access to the data. If authors do not respond, are not able or not willing to share their data, we will compare these studies with the included ones in terms of design, participants, intervention and quality.

### Risk of bias assessment

The validity of the included studies will be assessed using four criteria from the Cochrane 'Risk of Bias' assessment tool.[77] This tool identifies possible sources of bias, including the adequate generation of allocation sequence, the allocation concealment, blinding of assessors and dealing with incomplete outcome data (this is assessed as positive when intention-to-treat analyses were conducted, meaning that all randomised participants were included in the analyses). Only data from published papers will be used to determine the risk of bias in order to use a consistent procedure across studies that does or does not share data. Two researchers will conduct the quality assessment independently and agreement rates will be reported. Disagreement will be solved through discussion.

### Missing data

IPD-MA will be conducted according to the intention-to-treat principle. Missing data are handled using a joint modelling approach to multiple imputation of IPD nested within studies.[78–80] In particular, we will use the R package jomo that uses Markov chain Monte Carlo techniques to draw replacements for the missing values.[81] This procedure is based on a multilevel imputation model that considers associations between continuous and categorical variables both at the level of participants (level 1) and studies (level 2). In addition, it allows for modelling between-study heterogeneity in the covariance matrices, which is especially useful when imputing variables that are completely missing from studies.[78] We will specify a multivariate empty imputation model including all available participant (level 1) and study (level 2) characteristics.[82] Assignment to intervention group versus control group will be used as a grouping variable in the imputation model to allow for treatment-specific intercept, variance and covariance parameters. Based on the final model, we will generate at least 20 imputed data sets. The number of burn-in iterations and the number of iterations between

imputed data sets will be chosen so that convergence can be ensured.[82] In the case of persistent convergence problems, we will reduce the number of model parameters by dropping predictors and/or imposing constraints to the model (eg, assuming a common level 1 covariance matrices across studies).

### Analysis

#### Conventional meta-analysis on study level

We will first conduct a conventional meta-analysis using data from the published papers. This will enable us to examine whether studies that did not provide data might bias the results of our IPD-MA. This will be done by comparing those studies contributed to the IPD data set with those who did not with regard to the outcomes, risk of bias and other study characteristics.

First, we will calculate the IRR for developing a depressive disorder in the intervention compared with the control group for each study based on published papers, and then pool the results using the Comprehensive Meta-Analysis Software package V.3. With regard to effects on depressive symptom severity, we will calculate Hedges' g as a measure of the effect size indicating the difference between the intervention and control conditions at post-treatment. These analyses will be done using a random effects DerSimonian-Laird model[83] because considerable heterogeneity between studies is expected. To test the homogeneity of effect sizes, we will calculate the $I^2$ statistic as an indicator of heterogeneity in percentages.[82] A value of 0%–40% indicates unimportant heterogeneity, and larger values indicate increasing heterogeneity, with 30%–60% as moderate, 50%–90% substantial and 75%–100% as considerable. We will calculate 95% CIs using the non-central $X^2$-based approach.[84] Small sample bias will be tested by inspecting the funnel plot visually, the Egger's test, and Duval and Tweedie's trim-and-fill procedure,[85] which yields an estimate of the effect size after small sample bias has been taken into account.[86]

#### Individual participant data meta-analysis

For the IPD-MA, we will use a one-step data analysis approach. This is currently the best possible meta-analysis approach with the standard two-step analysis being at best equivalent in some scenarios.[87] All models are repeated for all of the defined follow-ups.

##### Effects on MDD onset

We will use multilevel logistic regression analysis based on the imputed data sets for predicting the occurrence of MDD, including the assignment to intervention group versus control group as the focal predictor. Patient-level data will be treated as level 1 and study-level data as level 2. Models will include both random intercepts and random slopes to capture both unobserved heterogeneity in trial populations (intercept) and trial effectiveness (slope). We will calculate ORs and corresponding 95% CIs, and then calculate the numbers needed to treat (NNT) and

corresponding 95% CIs in order to avoid one additional MDD. In addition, we will conduct two additional analyses taking varying observation periods and time to MDD onset explicitly into account. To control for differences in observation periods, we will use a multilevel binomial regression analysis with a complementary log-log link and offset for time since baseline, which provides an estimate of the treatment effect in terms of the IRR for developing an MDD.[88] To assess the differences in time to MDD onset, we will use multilevel Cox proportional hazards models, which provide an estimate of the treatment effect in terms of the HR for developing an MDD.

### Effects on symptom severity

We will predict standardised depressive symptom severity scores from intervention group versus control group and control for baseline depressive symptom severity using a multilevel linear regression analysis. Again, we will include both a random intercept and a random slope for treatment effects to capture both unobserved heterogeneity between study populations (intercept) and study effectiveness (slope). Hedges' g will be calculated as an effect size measure. The same approach will be used for analysing effects on other continuous outcome measures including quality of life and anxiety and suicidal ideation.

### Effects on response, remission and symptom deterioration

The standard criterion for measuring response in psychotherapy outcome research for depression is a 50% reduction on a standardised depression measure.[89] However, it can be argued that in individuals with subclinical symptoms a relative reduction of 50% of symptoms might be clinically less meaningful compared with individuals with major depression. Hence, we will additionally calculate response using a predefined absolute reduction in symptoms using the Reliable Change Index.[40] Remission will be defined using standard cut-off scores of the respective instruments. Symptom deterioration will be defined using a predefined absolute worsening of symptoms from baseline to follow-up using the Reliable Change Index[40] and 50% symptom increase. Generally, event occurrence will be predicted from treatment group using multilevel logistic regression analysis. We will proceed to calculate the OR and its 95% CIs, and then calculate the NNT and its 95% CIs in order to achieve one additional response, respectively remission as compared with the control group.[90]

### Moderator analyses

We will explore predictors of outcome (ie, prognostic variables) and moderators of the intervention effect (ie, prescriptive variables) by including selected participant-level and study-level variables as well as their interactions with the intervention as additional predictors in the multilevel (logistic) regression analyses. These analyses will be based on the total sample (ie, on the imputed data sets including all studies) and focus on predicting onset of MDD, depressive symptom severity

and symptom deterioration. Variables will be selected based on the combination of multiple criteria, including the amount of available/missing data, the bivariate associations with outcome measures in the intervention group and control group, and the convergence of the multiple imputation model. In order to increase statistical power, moderator analyses on long-term effects will be done using combined follow-up assessments to include all studies that contribute follow-up data.

### Subgroup analyses

We also plan to examine the effectiveness of the interventions and moderators of treatment outcome in subgroups that are of special interest for tailoring prevention programmes (eg, older adults, low-educated adults, minority status, mothers of newborns, medical conditions and individuals without lifetime history of depression). These analyses will be based on subsamples. Note that it will be necessary to generate new imputed data sets for these analyses to ensure congeniality with the imputation model.[80] The same strategy will be applied to investigate effects and moderators in specific intervention delivery forms (eg, internet, guided/unguided self-help, group format). However, whether these and other analyses in subgroups of interest should be conducted depends on the number of studies/participants that are eligible.

### Sensitivity analyses

A number of sensitivity analyses will be conducted in order to test the robustness of our findings. For example, we will run a separate model in which we exclude trials with high risk of bias. If a sufficient number of studies include the same outcome measurement (eg, for depressive symptom severity), we will conduct separate analyses using only this specific outcome measurement, instead of using the standardised score. We will also run a complete case analysis and compare the results with the intention-to-treat analysis in order to determine whether a difference exists between those that dropped out from the trials compared with those who persisted. Other sensitivity analyses may be necessary and will be decided on after all data have been collected and examined.

## DISCUSSION

The burden attributable to major depression is immense and although effective treatments are available, effects on disease burden are limited. Treatments so far failed to show that the prevalence of depression in the population can be reduced, even in those countries in which evidence-based treatments have been made widely available. Hence, new approaches are needed to reduce the burden of MDD at population level. This study will provide a precise estimate of the effects of indicated preventive interventions for subclinical symptoms of depression on short-term and long-term depressive symptom severity, MDD onset and other relevant outcome criteria. Using an IPD meta-analytic approach, we will be able to estimate

specific effects in relevant subgroups of interest and test whether the effectiveness depends on individual participant criteria.

Such approaches have been used with some frequency in medicine, but are less often applied in the field of psychological treatment outcome research, although recently a number of studies have been published[91–97] or are in preparation.[98–100] As the field moves towards personalised medicine, it is crucial to know specific effects for specific kinds of treatments for patients with certain characteristics in order to select the best possible treatment for an individual patient. IPD-MA allows to do this with sufficient statistical power.

However, such an approach also has a number of challenges. First, until such a study is published, it is very likely that the search is already outdated and more trials have already been published that could theoretically have been included. This is due to the fact that solely the process of obtaining and integrating primary data into one data set takes very long. Updating the search and including additional data sets within the review process needs to be balanced to what can be gained by doing so with regard to the specific research question investigated, as theoretically this process could be done repeatedly. For example, if effects in relevant investigated subgroups are consistent across trials, heterogeneity is low, the number of included studies and participants is reasonable, effects are clinically meaningful with narrow CIs for effect sizes, then it is unlikely that the inclusion of an additional study would result in meaningful changes that would justify the delay in publishing the results to be available for the scientific community and policymakers. On the other hand, if differences of effect sizes between specific subgroups are substantial, but moderator analyses are underpowered to detect such a difference and the inclusion of additional studies would change this, the additional value of updating the data set would potentially outweigh the disadvantages. Second, a limitation of the IPD-MA approach is that one very much relies on the variables that have been assessed in the primary study. In addition, many relevant predictors and moderators associated with depression onset or differential treatment response in the literature, such as, for example, lifetime history of depression, childhood adversities are not included in many of the published studies. However, recent advantages in statistics allow to account for between-study heterogeneity when imputing missing values and to impute variables that are systematically missing in studies.[78 101] Nevertheless, we argue that authors should include variables in primary studies that might eventually explain heterogeneity of treatment effects, even when the study is not powered to reliably investigate differential treatment effects. This would allow using these data in IPD-MA studies and might bring the field of precision medicine in psychological treatment outcome research substantially forward. Third, another challenge with IPD-MA is that often not all available trials can be included in the data set due to author non-response, lack of ethical approval to share the data or that data are not available anymore. This might introduce some bias, which is being addressed by comparing IPD findings with those of traditional meta-analyses in the present study.

## Ethics and dissemination

This paper is a study protocol for an individual patient data meta-analysis and does not require ethical approval. The investigators of the primary trials have obtained ethical approval for the data used in the present study and for sharing the data, if this was necessary, according to local requirements and was not covered from the initial ethic assessment. Only anonymised data are included in the data set which does not allow the identification of individual trial participants. Anonymised data collected are managed by CB and JAR and will be available for the complete research team. External research can request access to the data set for secondary analyses after publication of the results specified in this protocol, if local requirement of the original data should allow this.

This study will summarise the available evidence on the short-form and long-term effectiveness of preventive psychological interventions for the treatment of subthreshold depression and prevention of MDD onset. Identification of subgroups of patients in which those interventions are most effective will guide the development of evidence-based personalised interventions for patients with subthreshold depression.

**Author affiliations**
[1]Department of Clinical Psychology and Psychotherapy, Friedrich-Alexander University Erlangen Nuremberg, Erlangen, Germany
[2]Institute of Psychology, Leuphana University of Luneburg, Luneburg, Germany
[3]Chair for Psychological Methods and Diagnostics, Psychologische Hochschule Berlin, Berlin, Germany
[4]Department of Clinical, Neuro and Developmental Psychology, EMGO+ Institute for Health and Care Research, VU University Amsterdam, Amsterdam, The Netherlands

**Contributors** DDE and PC conceptualised and designed the study. PC contacted the primary authors. JAR and CB are responsible for building the database. JZ is responsible for data analyses. DDE drafted the manuscript and is the guarantor of the review. All authors critically revised the manuscript, and read and approved the final version.

**Funding** This research received no specific grant from any funding agency in the public, commercial or not-for-profit sectors. We acknowledge support by Deutsche Forschungsgemeinschaft and Friedrich-Alexander-Universität Erlangen-Nürnberg (FAU) within the funding programme Open Access Publishing.

**Competing interests** None declared.

**Patient consent** Not required.

**Ethics approval** This paper is a study protocol for an individual patient data meta-analysis and does not require ethical approval.

**Provenance and peer review** Not commissioned; externally peer reviewed.

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
