## [Reviewer comments · BMJ Open]

ARTICLE DETAILS

TITLE (PROVISIONAL)	Efficacy and moderators of psychological interventions in treating subclinical symptoms of depression and preventing Major Depressive Disorder onsets: Protocol for an individual patient data meta-analysis of randomized controlled trials
AUTHORS	Ebert, David; Buntrock, Claudia; Reins, Jo Annika; Zimmermann, Johannes; Cuijpers, Pim

VERSION 1 – REVIEW

REVIEWER	Cindy Hagan Caltech, USA
REVIEW RETURNED	25-Jul-2017

GENERAL COMMENTS	General Summary: This paper presents an outline for conducting an individual patient data meta-analysis of randomized controlled trials to examine whether the treatment of subclinical symptoms of depression prevents relapse and promotes recovery from depression. The paper also seeks to identify important moderators of psychological interventions for depression in an effort to facilitate the overall goal of treatment being tailored to an individual or groups of individuals. I am slightly concerned that there may be too few RCTs that have examined subclinical depression to warrant meta-analysis. Nonetheless, the authors should be commended for their pursuit of answers to such a worthy research topic. Minor points for consideration: -There are minor grammatical errors throughout the paper which will need to be resolved prior to publication.-The abstract states that ethical consent has been obtained, but it states within the paper that ethical permission is not needed for the study to be conducted. More clarity is needed with respect to whether ethical permission should be obtained and documented prior to compiling data for the study. Authors should consider how anonymity of data may differ across countries and list the steps they will take to ensure for data to remain anonymous.-As this is a study protocol, a Results section should be created with "N/A" underneath, as per journal recommendations. Similarly, a date range for when the meta-analysis will be conducted should be supplied (e.g., Jan 2018-Dec 2018).
--

	-Authors refer to "prevention of Major Depression" throughout the paper, which can be misconstrued as long-term prevention of MDD, which would require longitudinal studies lasting several years and goes beyond the scope of the meta-analysis. It may be better to rephrase these statements as "prevention of Major Depression in the short term" or "prevention of MDD onsets" to avoid such potential misinterpretation. -Authors mention that analyses will be conducted where sufficient data are available across the pooled studies. Some guidelines as to what constitutes "sufficient data" could be provided (e.g., minimum number of cases needed to conduct adequately powered analyses).
--	--

REVIEWER	Juan Bellón IBIMA, redIAPP and SAS; Department of Public Health and Psychiatry, University of Málaga, Spain.
REVIEW RETURNED	04-Aug-2017

GENERAL COMMENTS	Thanks for letting me review this interesting protocol. It aims to examine the short and long-term effects of psychological interventions compared to control groups in adults with subthreshold depression on several outcomes (depression symptom severity, treatment response, remission, deterioration, quality of life, anxiety, and the prevention of MDD onsets) as well as to explore some moderators on individual patient- and study level using an individual-patient data meta-analysis approach. The protocol is well written and clear, the study design is appropriate to answer the research questions. The references are up-to-date and appropriate. The methods are also pertinent and sufficiently described and limitations and potential bias are addressed correctly. To the best of my knowledge this protocol is free from concerns over publication ethics. With the intention of improving, if possible, this protocol, here are some comments:  • I have not had access to the PRISMA-P check-list [Shmseer et al. Preferred reporting items for systematic review and meta-analysis protocols (PRISMA-P) 2015: elaboration and explanation. BMJ. 2015 Jan 2;349:g7647] completed by the authors, although most of the items on this check-list are included in this protocol. I have missed in the text of the protocol some reference to the item 17: "Describe how the strength of the body of evidence will be assessed (such as GRADE)". • In the introduction (Page 5, lines 44-46) the authors say "However, four studies using clinician-rated outcomes did not indicate significant positive results." Please, indicate the reference or references for this statement. • Page 10 (lines 48-51), the authors say "A value of 0% indicates no observed heterogeneity, and larger values indicate increasing heterogeneity, with 25% as low, 50% as moderate, and 75% as high". Please, Update these parameters according to the Cochrane HandBook (0-40% might be unimportant heterogeneity, 30-60% moderate, 50-90% substantial and 75-100% considerable) • Page 11 (line 34), the authors say "Effects on symptom severity: We will use a multilevel regression analysis". I think is more appropriate "Effects on symptom severity: We will use a multilevel linear regression analysis"
--

REVIEWER	Łukasz Gawęda 1. Department of Psychiatry and Psychotherapy, University Medical Center Hamburg Eppendorf, Hamburg, Germany 2. II Department of Psychiatry, Medical University of Warsaw, Poland
REVIEW RETURNED	31-Aug-2017

GENERAL COMMENTS	The authors presented a comprehensive protocol of a meta-analysis of studies on psychological interventions in subthreshold depression. The main rationale of the study is to investigate the role of several important mediators of potential psychotherapeutic changes. The protocol caught my attention as I am convinced that looking for specific mediators of change is of great importance for clinical practice, as it may have an impact on further developments of psychotherapeutic interventions for depressive patients. I have no major methodological concerns to this piece. Introduction is well written with a good structure and the rationale of the meta-analysis is clearly presented. The authors used gold-standard statistical methods, which are well explained. Limitations are considered in the discussion section. Please see some minor comments/questions that may help to improve some parts of the protocol:  1. In terms of inclusion criteria, I suggest that the authors specify whether they are focused on subthreshold depressive symptoms as a primary “diagnosis” and exclude patient with co-morbid diagnosis (e.g. anxiety disorders, substance abuse etc.). Otherwise, it should be specified if and how co-morbidity will be considered in the meta-analysis. 2. There is increasing number of on-line studies for patients with depression. The criteria may suggest that these are also included (psychological interventions in general), but I suggest to clear this issue in the paper. 3. The term ‘psychological intervention’ is very general term (interpersonal therapy, psychodynamic therapy, psychoeducation etc.) and thus may be somehow misleading for readers. I suggest to define it in the paper. 4. The authors suggested very comprehensive analysis of moderators, which includes more than twenty variables. As they claimed “It is expected that not all studies that will be included assessed all variables. Hence, variables will only be examined if sufficient data across studies are available.” Please clear what is meant by sufficient data? 5. Given that suicidality (suicidal thoughts, suicidal attempts) is an important clinical outcome in the treatment of depressive symptoms I am interested in whether the authors considered it to include in their meta-analysis protocol? To sum up, by considering the effect of different moderators on the effectiveness of psychological treatments for subthreshold depression this piece may provide valuable clinical information for further development of evidence-based interventions. Hence, I recommend the protocol for publication after some minor issues are considered.
--

VERSION 1 – AUTHOR RESPONSE

Reviewer: 1

This paper presents an outline for conducting an individual patient data meta-analysis of randomized controlled trials to examine whether the treatment of subclinical symptoms of depression prevents relapse and promotes recovery from depression. The paper also seeks to identify important moderators of psychological interventions for depression in an effort to facilitate the overall goal of treatment being tailored to an individual or groups of individuals.

Reviewer 1, comment1: I am slightly concerned that there may be too few RCTs that have examined subclinical depression to warrant meta-analysis. Nonetheless, the authors should be commended for their pursuit of answers to such a worthy research topic.

Authors' response: We thank the reviewer for the thorough assessment of our manuscript. There is already a published meta-analysis on study level on the treatment of subclinical depression, since the search for this study has been conducted many new trials have been conducted. Hence, we're optimistic that we will gain sufficient data to conduct a meta-analyses.

Cuijpers, P., Koole, S. L., van Dijke, A., Roca, M., Li, J., & Reynolds 3rd, C. F. (2014). Psychotherapy for subclinical depression: meta-analysis. *Br J Psychiatry*, 205(4), 268–274.
<http://doi.org/10.1192/bjp.bp.113.138784>

Minor points for consideration:

Reviewer 1, comment 2: There are minor grammatical errors throughout the paper which will need to be resolved prior to publication.

Authors' response: We thank the reviewer for the hint and now thoroughly checked the MS for grammatical errors.

Reviewer 1, comment 3: The abstract states that ethical consent has been obtained, but it states within the paper that ethical permission is not needed for the study to be conducted. More clarity is needed with respect to whether ethical permission should be obtained and documented prior to compiling data for the study. Authors should consider how anonymity of data may differ across countries and list the steps they will take to ensure for data to remain anonymous.

Authors' response: We now clarified that the investigators of the primary trials, that are included in the dataset have obtained ethical permission to use the data.

Changes in the MS, abstract: The investigators of the primary trials have obtained ethical approval for the data used in the present study and for sharing the data, if this was necessary according to local requirements and was not covered from the initial ethic assessment.

Reviewer 1, comment 4: As this is a study protocol, a Results section should be created with "N/A" underneath, as per journal recommendations. Similarly, a date range for when the meta-analysis will be conducted should be supplied (e.g., Jan 2018-Dec 2018).

Authors' response: We now included a date range for when the IPD meta-analysis will be conducted in the revised MS. We did not find in the instructions for authors for study protocol the information that a result section with "N/A" underneath should be created in study protocol.

Changes in the MS, page 9: (studies published up to December 2017 will be considered for inclusion

Reviewer 1, comment 5: Authors refer to "prevention of Major Depression" throughout the paper, which can be misconstrued as long-term prevention of MDD, which would require longitudinal studies lasting several years and goes beyond the scope of the meta-analysis. It may be better to rephrase these statements as "prevention of Major Depression in the short term" or "prevention of MDD onsets" to avoid such potential misinterpretation.

Authors' response: We now changed "prevention of Major Depression" to "prevention of MDD onsets" throughout the MS.

Reviewer 1, comment 6: Authors mention that analyses will be conducted where sufficient data are available across the pooled studies. Some guidelines as to what constitutes "sufficient data" could be provided (e.g., minimum number of cases needed to conduct adequately powered analyses).

Authors' response: Thank you for alerting us to this lack of precision. After considering several potential solutions, we realized that the selection of predictors in an IPD-MA is very complex and should not be solely guided by the number of available cases. For example, the power to detect predictor-treatment interactions does not only depend on the effect size in the population, number of cases, and alpha error, but also on the heterogeneity of the effect between studies. Moreover, although multiple imputation is able to handle predictors with a substantial amount of missing values, including such predictors can lead to convergence problems depending on covariation with, and missingness of, further predictors in the model. Thus, we finally refrained from providing specific guidelines for what constitutes "sufficient data" and now explicitly state that we will take multiple criteria into account when selecting predictors.

Changes in the MS: Variables will be selected based on the combination of multiple criteria, including the amount of available/missing data, the bivariate associations with outcome measures in the intervention- and control-group, and the convergence of the multiple imputation model.

Reviewer: 2

Thanks for letting me review this interesting protocol. It aims to examine the short and long-term effects of psychological interventions compared to control groups in adults with subthreshold depression on several outcomes (depression symptom severity, treatment response, remission, deterioration, quality of life, anxiety, and the prevention of MDD onsets) as well as to explore some moderators on individual patient- and study level using an individual-patient data meta-analysis approach. The protocol is well written and clear, the study design is appropriate to answer the research questions. The references are up-to-date and appropriate. The methods are also pertinent and sufficiently described and limitations and potential bias are addressed correctly. To the best of my knowledge this protocol is free from concerns over publication ethics. With the intention of improving, if possible, this protocol, here are some comments:

Authors' response: We thank Reviewer 2 for the positive assessment of our work.

Reviewer 2, comment 1: I have not had access to the PRISMA-P check-list [Shmseer et al. Preferred reporting items for systematic review and meta-analysis protocols (PRISMA-P) 2015: elaboration and explanation. *BMJ*. 2015 Jan 2;349:g7647] completed by the authors, although most of the items on this check-list are included in this protocol. I have missed in the text of the protocol some reference to the item 17: "Describe how the strength of the body of evidence will be assessed (such as GRADE)". <https://www.ncbi.nlm.nih.gov/pmc/articles/PMC428525/>

Authors' response: We now included the PRISMA statement in the revised version.
Changes in the MS:

Reviewer 2, comment 2: In the introduction (Page 5, lines 44-46) the authors say “However, four studies using clinician-rated outcomes did not indicate significant positive results.” Please, indicate the reference or references for this statement.

Authors' response: We now included the corresponding references in the revised MS.

Changes in the MS: However, four studies using clinician-rated outcomes did not indicate significant positive results (Cuijpers et al., 2014).

Cuijpers, P., Koole, S. L., van Dijke, A., Roca, M., Li, J., & Reynolds 3rd, C. F. (2014). Psychotherapy for subclinical depression: meta-analysis. *Br J Psychiatry*, 205(4), 268–274.

<http://doi.org/10.1192/bjp.bp.113.138784>

Reviewer 2, comment 3: Page 10 (lines 48-51), the authors say “A value of 0% indicates no observed heterogeneity, and larger values indicate increasing heterogeneity, with 25% as low, 50% as moderate, and 75% as high”. Please, Update these parameters according to the Cochrane HandBook (0-40% might be unimportant heterogeneity, 30-60% moderate, 50-90% substantial and 75-100% considerable)

Authors' response: We now updated the values indicating heterogeneity according to the Cochrane Handbook.

Changes in the MS, page 10: A value of 0-40% indicates unimportant heterogeneity, and larger values indicate increasing heterogeneity, with 30-60% as moderate, 50-90% substantial and 75-100% as considerable.[75]

Reviewer 2, comment 4: Page 11 (line 34), the authors say “Effects on symptom severity: We will use a multilevel regression analysis”. I think is more appropriate “Effects on symptom severity: We will use a multilevel linear regression analysis”

Authors' response: We changed the sentence as suggested.

Changes in the MS:, page 11: We will use a multilevel linear regression analysis predicting standardised depression severity scores from treatment group and controlling for baseline depression severity.

Please leave your comments for the authors below The authors presented a comprehensive protocol of a meta-analysis of studies on psychological interventions in subthreshold depression. The main rationale of the study is to investigate the role of several important mediators of potential psychotherapeutic changes. The protocol caught my attention as I am convinced that looking for specific mediators of change is of great importance for clinical practice, as it may have an impact on further developments of psychotherapeutic interventions for depressive patients.

I have no major methodological concerns to this piece.

Introduction is well written with a good structure and the rationale of the meta-analysis is clearly presented. The authors used gold-standard statistical methods, which are well explained. Limitations are considered in the discussion section.

Please see some minor comments/questions that may help to improve some parts of the protocol:

Author's response. We also thank reviewer three for reviewing our manuscript and the positive assessment of our work.

Reviewer 3, comment 1: In terms of inclusion criteria, I suggest that the authors specify whether they are focused on subthreshold depressive symptoms as a primary “diagnosis” and exclude patient with co-morbid diagnosis (e.g. anxiety disorders, substance abuse etc.). Otherwise, it should be specified if and how co-morbidity will be considered in the meta-analysis.

Authors' response: We do not exclude comorbidity as a diagnoses. Comorbidity will be considered as a potential effect modifier.

This is outlined on page 8 of our manuscript "Clinical and personality characteristics that will be investigated, if sufficiently available, include depressive symptom severity,[49] lifetime-history of MDD,[50,51] number of previous depressive episodes,[50,52] anxiety symptoms,[50] comorbid mental health disorder (e.g. anxiety disorder)[51], previous exposure to depression treatment, family history of common mental health disorders,[51,53,54] global assessment of functioning, sleeping problems,[55–57] neuroticism,[49] recent life stress,[58] childhood adversities,[54] traumatic events,[59] significant life events (in the previous year),[60,61], daily hassles, emotion regulation,[62] poor self-perceived health (quality of life),[50,55,61] self-esteem,[63–65] (chronic) medical conditions,[56,57,66] physical functioning/ disability,[55] mastery, worrying, Body-Mass-Index, rumination, interpersonal problems,[52,61] body dissatisfaction,[65,67] physical activity level,[55,68] diet quality,[68] alcohol / substance use,[51,55,61] smoking,[55,66] resilience,[69] social support/ integration,[51,56,62,65] perceived social rejection/ mobbing."

Reviewer 3, comment 2: There is increasing number of on-line studies for patients with depression. The criteria may suggest that these are also included (psychological interventions in general), but I suggest to clear this issue in the paper.

Authors' response: Yes, we consider also online-delivered psychological interventions suitable for inclusion and now specify this more clearly in the manuscript

Changes in the MS, page 6: In this IPD MA, we will a) include randomized trials in which b) the effects of a psychological treatment (delivered individually, in a group-, bibliotherapy, internet-based format) were compared with a comparison group (waiting list, care-as-usual, psychological placebo, pill placebo, antidepressant medication)

Reviewer 3, comment 3: The term 'psychological intervention' is very general term (interpersonal therapy, psychodynamic therapy, psychoeducation etc.) and thus may be somehow misleading for readers. I suggest to define it in the paper.

Authors' response: Following the reviewers suggestion, we now include a definition of psychological intervention.

Changes in the MS, page 6: Psychological interventions were defined as "application of psychological mechanisms and interpersonal stances derived from psychological principles for the purpose of assisting people to modify their behaviors, cognitions, emotions, and/or other personal characteristics in directions that the participants deem desirable"

Reviewer 3, comment 4: The authors suggested very comprehensive analysis of moderators, which includes more than twenty variables. As they claimed "It is expected that not all studies that will be included assessed all variables. Hence, variables will only be examined if sufficient data across studies are available." Please clear what is meant by sufficient data?

Authors' response: see our response to Reviewer 1, comment 6.

Changes in the MS: Variables will be selected based on the combination of multiple criteria, including the amount of available/missing data, the bivariate associations with outcome measures in the intervention- and control-group, and the convergence of the multiple imputation model.

Reviewer 3, comment 5: Given that suicidality (suicidal thoughts, suicidal attempts) is an important clinical outcome in the treatment of depressive symptoms I am interested in whether the authors considered it to include in their meta-analysis protocol?

Authors' response: We fully agree with the reviewer that this is an important outcome and now included it as additional outcome criteria

Changes in the MS, abstract / page 7: "We will use the following types of outcome criteria: a) onset of major depression, b) time to major depression onset, c) observer- and self-reported depressive symptom severity, d) response, e) remission, f) symptom deterioration, g) quality of life, and h) anxiety i) suicidal thoughts and behavior."

Comment: To sum up, by considering the effect of different moderators on the effectiveness of psychological treatments for subthreshold depression this piece may provide valuable clinical information for further development of evidence-based interventions. Hence, I recommend the protocol for publication after some minor issues are considered.

Authors' response: thanks!

VERSION 2 – REVIEW

REVIEWER	Cindy Hagan California Institute of Technology, USA
REVIEW RETURNED	13-Dec-2017

GENERAL COMMENTS	Thank you for resolving the issues that were raised.
--

REVIEWER	Juan A. Bellon El Palo Health Center. redIAPP. IBIMA. Department of Public Health and Psychiatry. University of Malaga.
REVIEW RETURNED	19-Dec-2017

GENERAL COMMENTS	The authors have responded adequately to all my comments. I look forward to seeing the results soon
---

REVIEWER	Lukasz Gaweda 1. Department of Psychiatry and Psychotherapy, Medical University Center, Hamburg-Eppendorf, Germany 2. II Department of Psychiatry, Medical University of Warsaw, Poland
REVIEW RETURNED	08-Jan-2018

GENERAL COMMENTS	The authors improved the manuscript. I do not have any further comments.
--